# ANYAVATAR: DYNAMIC AND CONSISTENT AUDIO-DRIVEN HUMAN ANIMATION FOR MULTIPLE CHARACTERS

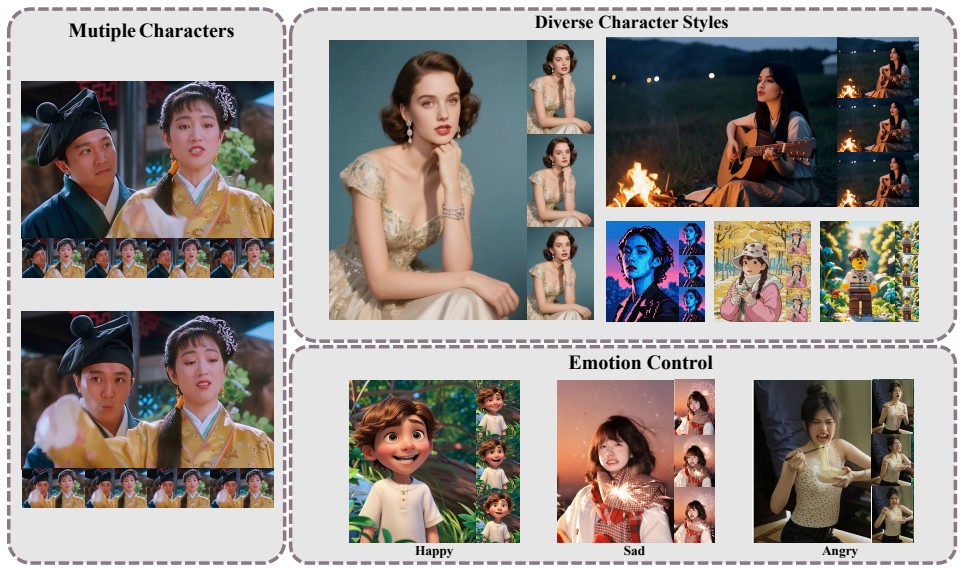

Figure 1: AnyAvatar can generate videos using a character image and audio as input. AnyAvatar enables the creation of multi-character, highly consistent, and dynamic human animations that accurately reflect the emotions expressed in the audio.

## ABSTRACT

Recent years have witnessed significant progress in audio-driven human animation. However, critical challenges remain in (i) generating highly dynamic videos while preserving character consistency, (ii) achieving precise emotion alignment between characters and audio, and (iii) enabling multi-character audio-driven animation. To address these challenges, we propose AnyAvatar, a multimodal diffusion transformer (MM-DiT)-based model capable of simultaneously generating dynamic, emotion-controllable, and multi-character dialogue videos. Concretely, AnyAvatar introduces three key innovations: (i) A character image injection module is designed to replace the conventional addition-based character conditioning scheme, eliminating the inherent condition mismatch between training and inference. This ensures the dynamic motion and strong character consistency; (ii) An Audio Emotion Module (AEM) is introduced to extract and transfer the emotional cues from an emotion reference image to the target generated video, enabling fine-grained and accurate emotion style control; (iii) A Face-Aware Audio Adapter (FAA) is proposed to isolate the audio-driven character with latent-level face mask, enabling independent audio injection via cross-attention for multi-character scenarios. These innovations empower AnyAvatar to surpass state-of-the-art methods on benchmark datasets and a newly proposed wild dataset, generating realistic avatars in dynamic, immersive scenarios. The source code and model weights will be released publicly.

# 1 INTRODUCTION

In recent years, Diffusion Transformers (DiT) have significantly advanced video generation. Among these developments, text-to-video and image-to-video techniques Bar-Tal et al. (2024); Zhou et al. (2024); Blattmann et al. (2023a;b); Guo et al. (2023); Zhou et al. (2022); Gupta et al. (2023); Wang et al. (2023); Ho et al. (2022); Brooks et al. (2022); Wang et al. (2020); Singer et al. (2022); Li et al. (2018); Villegas et al. (2022); Lin et al. (2025b) have gained increasing attention due to their near-practical applicability. Audio-driven human animation, has experienced explosive growth, as it enables realistic human video synthesis with minimal input. Recent DiT-based approaches Zhang et al. (2023); Wang et al. (2024a); Meng et al. (2024); Cui et al. (2024); Lin et al. (2025a) have demonstrated superior performance in audio-driven generation compared to existing methods.

Current audio-driven human animation methods can be broadly categorized into two paradigms: portrait animation and full-body animation. Portrait animation methods Wang et al. (2024a); Meng et al. (2024); Cui et al. (2024) focus exclusively on facial movements while maintaining static or simplistic backgrounds. Full-body animation methods Lin et al. (2025a); Tu et al. (2025); Gan et al. (2025); Kong et al. (2025); Gao et al. (2025) address this spatial limitation by extending motion to the full body. However, they face persistent challenges including unnatural character movements, misalignment between audio emotions and facial expressions, and an inability to drive multi-character scenes with audio. These limitations currently represent the most significant barrier to developing truly convincing audio-driven human animations.

Recent advances in audio-driven human animation have achieved significant progress, yet critical challenges persist in motion quality, character consistency, emotion alignment, and multi-character audio-driving. For instance, Hallo-3 Cui et al. (2024), a DiT-based portrait animation method, generates only facial movements while neglecting body motion. OmniHuman-1 Lin et al. (2025a) introduces a multimodal motion-conditioned hybrid training strategy to mitigate data scarcity issues. OmniAvatar Gan et al. (2025) proposes an audio-driven human animation model trained with LoRA. WanS2V Gao et al. (2025) proposes an efficient model that integrates text and audio control, enabling high-quality and stable audio-driven video generation of characters in complex cinematic scenes. StableAvatar Tu et al. (2025) combines a timestep-aware audio adapter and a dynamic weighted sliding-window strategy to achieve infinite-length, identity-consistent video synthesis. However, its approach greatly limits the dynamic movements. These limitations underscore the need for more robust solutions. To address these gaps, our work focuses on three key objectives: (i) improving dynamic expressiveness while preserving character identity, (ii) ensuring precise emotion synchronization between audio and video, and (iii) enabling realistic multi-character dialogue generation for real-world applications.

First, current audio-driven human animation methods typically rely on reference images during inference to enforce consistency between the generated video and the reference. However, this approach often leads to unnatural motion, as the model tends to replicate expressions and poses from the reference rather than generating dynamic, audio-aligned movements. To overcome this limitation, we propose a character image injection module, which transforms human image features into representations more amenable to model learning. By injecting these features along the channel dimension, we avoid the trade-off between dynamism and consistency that arises from direct latent space usage, ensuring coherence between training and inference.

Second, we introduce an Audio Emotion Module (AEM) to align video characters' emotions with those conveyed in the audio. This module leverages reference images to guide emotion mapping, ensuring that facial expressions accurately reflect the audio's affective content, thereby improving realism in human animation.

Finally, to address the challenge of multi-character animation, we propose a Face-Aware Audio Adapter (FAA). This module applies a face mask to latent features extracted from the input, generating face-masked video latents that are then fused with audio information. Since the audio primarily influences the masked face region, we can independently drive different characters using distinct audio inputs, enabling realistic multi-character dialogue generation for cinematic applications.

Extensive experiments demonstrate that our framework effectively drives multi-person scenarios with audio, significantly improving both dynamism and consistency. Our key contributions are as follows:

- A character image injection module that resolves the dynamism-consistency trade-off caused by reference image usage, enhancing overall motion quality in foreground and background.

- An Audio Emotion Module (AEM) that aligns video characters' emotions with audio-driven affective cues, improving realism in facial expressions.

- A Face-Aware Audio Adapter (FAA) that enables localized audio-driven animation for multiple characters by masking targeted face regions in the latent space, facilitating multi-character dialogue generation.

## 2 RELATED WORK

**Audio-conditioned portrait animation.** Hallo Xu et al. (2024) proposes an innovative hierarchical audio-driven visual synthesis approach based on diffusion models, which integrates generative models, denoisers, temporal alignment techniques, and a reference network to achieve precise synchronization between audio inputs and visual outputs. V-Express Wang et al. (2024a) balances strong and weak control signals through progressive drop operations, enabling effective use of weak signals like audio in portrait video generation. EchoMimic Meng et al. (2024) innovatively uses both audio and facial landmarks for training, addressing the instability and unnatural results of using audio or landmarks alone, enabling the generation of more natural portrait videos. Loopy Jiang et al. (2024) learns natural motion and improves audio-portrait movement correlation through designed temporal modules and an audio-to-latents module, eliminating the need for manual motion templates to generate more realistic and high-quality videos. Hallo3 Cui et al. (2024) is designed with a Transformer-based identity reference network to ensure facial identity consistency, and explores speech audio conditions and motion frame mechanisms to enable the model's audio-driven capabilities.

**Audio-conditioned full-body animation.** In OmniHuman-1 Lin et al. (2025a), a multimodal motion condition hybrid training strategy is introduced, enabling the model to benefit from data augmentation with mixed conditions, thereby overcoming the challenges faced by previous methods due to the scarcity of high-quality data. StableAvatar Tu et al. (2025) integrates a timestep-aware audio adapter and a dynamically weighted sliding window strategy, enabling infinite-length video synthesis while addressing issues of audio synchronization and segment drift. Its innovative audio local guidance mechanism further enhances the naturalness of audio-driven expressions and movements. OmniAvatar Gan et al. (2025) proposes an audio-conditioned full-body avatar video generation model trained with LoRA. By employing multi-level, pixel-wise audio embeddings, it effectively enhances natural and adaptive body movements as well as high-precision lip synchronization. MultiTalk Kong et al. (2025), a novel framework for audio-driven multi-person video generation, which introduces multi-stream audio injection and Label Rotary Position Embedding to address audio-person binding, along with innovative training strategies, effectively enabling instruction-following and dynamic, realistic multi-person video synthesis.WanS2V Gao et al. (2025) proposes a high-quality audio-driven human animation method that combines global text-based control with fine-grained audio-driven motion, supports complex multi-person scenarios and stable long video generation.

## 3 METHODS

Given a reference image, a driving audio, and a facial mask of the character, our method can generate talking videos of single or multiple characters based on the driving audio. The overall framework of our method is illustrated in the figure 2. Specifically, we adopt HunyuanVideo Kong et al. (2024) as our backbone. It is a video generation model built upon the MM-DiT architecture. In Section 3.1, we explore a character image injection module, which can maintain both character consistency and vividness. Then, in Section 3.2, we discuss how to apply an audio adapter to face region to enable multi-character audio-driven animation. In Section 3.3 we discuss an emotion alignment module.

### 3.1 CHARACTER IMAGE INJECTION MODULE

In previous I2V methods, padding frames were often used for video inference. While this approach ensures good integrity and consistency of characters, backgrounds, and foregrounds, it also limits the motion dynamics of the generated video. Additionally, padding frames can lead to misalignment between the training and inference processes. Removing padding frames for video inference results

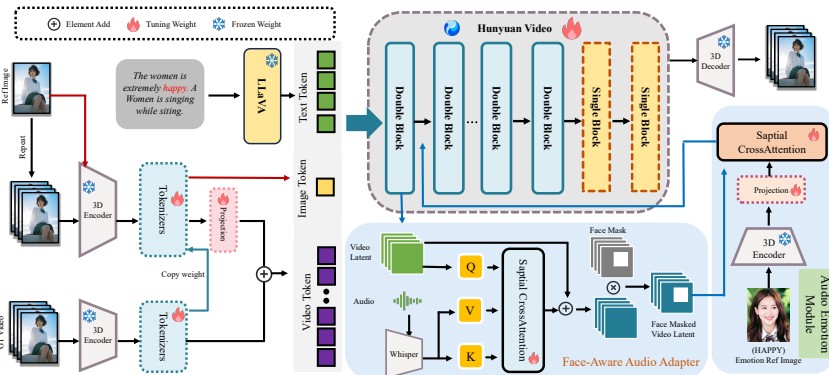

Figure 2: The framework of AnyAvatar. It consists of three parts: (1) Character Image Injection Module, which ensures high consistency of the character while maintaining high dynamics; (2) Audio Emotion Module, which aligns the character's facial expressions in the video with the emotions in the audio; and (3) Face-aware Audio Adapter, which enables audio-driven multiple characters.

in better motion dynamics but severely compromises character consistency and integrity. Therefore, we explored three character image injection mechanisms, as illustrated in Figure 3: (a) the reference image and video are processed through the same tokenizer, and the generated latents are concatenated in the token dimension; (b) the character image is first repeated T times (T represents the length of the video) and concatenated with the original video in the channel dimension, then fed into tokenizer1, while the character reference image is fed into tokenizer2, and both are concatenated in the token dimension before fed to the model; (c) the reference image is first repeated T times and fed into tokenizer2, then added directly to the video latent through a projection module composed of fully connected layers fed to the model. The mechanism (c) shows better results compared to mechanisms (a) and (b), as it improves the dynamics of motion while ensuring the consistency and integrity of characters, backgrounds, and foregrounds in the video, significantly enhancing video quality. For specific ablation comparisons experience, please refer to the experiment section. Since the backbone's tokenizer1 is specifically trained for video, we need to add an extra tokenizer2 to fit the image branch. The weights of this tokenizer are copied from the backbone's tokenizers, and we found that this approach accelerates model convergence.

## 3.2 FACE-AWARE AUDIO ADAPTER

In terms of audio conditioning, we use Whisper Radford et al. (2023) for audio feature extraction, and for face masks, we employ the InsightFace Ren et al. (2023) method to detect the bounding box of the facial region. Given an audio-video sequence consisting of $n'$ frames, we extract audio features for each frame, yielding a feature of shape $n' \times 10 \times d$, where 10 denotes the number of tokens per audio frame. The corresponding video latent representations are temporally compressed by a pretrained 3D VAE into $n$ frames, with $n = \left\lfloor \frac{n'}{4} \right\rfloor + 1$, where the additional 1 accounts for the initial, uncompressed frame, and 4 is the temporal compression ratio. Furthermore, to incorporate identity information, an identity image is concatenated at the beginning, resulting in a video latent of $n + 1$ frames.

To ensure temporal alignment between the audio features and the compressed video latent, we first pad the audio feature sequence prior to the initial frame, producing a total of $(n + 1) \times 4$ audio frames. We then aggregate every four consecutive audio frames into one, resulting in a temporally aligned audio feature tensor $g_A$ that matches the structure of the video latent representation. To ensure temporal alignment between the face mask and the compressed video latent, we set the face mask corresponding to the initial frame to 1, and also make it contain a total of $(n + 1) \times 4$ mask frames. This results in a mask $g_M$ that is both temporally and spatially aligned with the video latent.

$$g_A = \text{Rearrange}(g_{A,0}) : [b, (n + 1) \times 4, 10, d] \rightarrow [b, (n + 1), 40, d]. \quad (1)$$

With the temporally aligned audio features $g_A$, we introduce audio information into the video latent representation $y_t$ using a cross-attention mechanism. To prevent interference across different time steps, we adopt a **spatial cross-attention** strategy that performs audio injection separately for each time step. Specifically, each audio frame interacts only with the spatial tokens of its temporally aligned video frame, and cross-attention is applied independently at each temporal index. To this

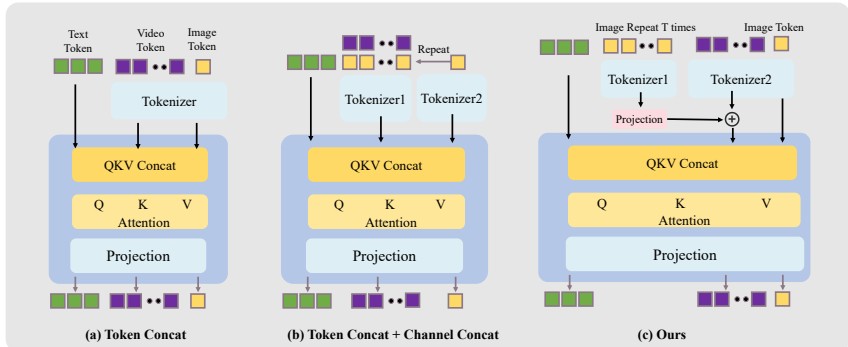

Figure 3: Three types of Character Image Injection Module.

end, we decouple the temporal dimension from the spatial dimensions of the video latent and apply attention solely along the spatial axes:

$$y'_{t,A} = \text{Rearrange}(y_t) : [b, (n+1)wh, d] \rightarrow [b, n+1, wh, d],$$
$$y''_{t,A} = y'_{t,A} + \alpha_A \times \text{CrossAttn}(g_A, y'_t) \times g_M, \quad (2)$$
$$y_{t,A} = \text{Rearrange}(y''_{t,A}) : [b, n+1, wh, d] \rightarrow [b, (n+1)wh, d],$$

where $\alpha_A$ is a weight to control the influence of the audio feature.

### 3.3 AUDIO EMOTION MODULE

To align the emotion conveyed in the audio with the character's facial expression, we compress the emotional reference image into features using a pretrained 3D VAE, and then inject these features into the **Double Block** of HunyuanVideo through an FC layer and **spatial cross-attention mechanism**. Specifically, the reference image features serve as the Key and Value, while the original video latent representation serves as the Query. This approach fuses information from the emotional reference image with the masked video latent $y_{t,A}$, enabling the model to better understand the relationship between audio emotion and facial expressions. To formalize this process, we first encode the emotional reference image $E_{\text{ref}} = \text{Encoder}(I_{\text{ref}})$, where $E_{\text{ref}}$ denotes the encoded feature of the emotional reference image $I_{\text{ref}}$. Next, to integrate these features into the video latent representation, we perform the following steps: We first reshape the video latent $y_{t,A}$ into temporal-spatial dimensions as $y'_{t,A}$, then apply an FC layer and spatial cross-attention to inject emotional features: $y''_{t,A,E}$, and finally restore the original structure:

$$y'_{t,A} = \text{Rearrange}(y_{t,A}) : [b, (n+1)wh, d] \rightarrow [b, n+1, wh, d],$$
$$y''_{t,A,E} = y'_{t,A} + \gamma_E \times \text{CrossAttn}(\text{FC}(E_{\text{ref}}), y'_{t,A}), \quad (3)$$
$$y_{t,A,E} = \text{Rearrange}(y'_{t,A,E}) : [b, n+1, wh, d] \rightarrow [b, (n+1)wh, d],$$

where $\gamma_E$ is a learnable scaling factor that controls the influence of the emotional reference features on the video latent. Notably, we found that inserting this module into a **Single Block** does not allow the model to effectively learn emotion. In contrast, integrating it into a **Double Block** enables the model to better drive character emotions. This suggests that the Double Block plays a crucial role in capturing and representing emotional details during complex emotion-to-expression mapping tasks.

### 3.4 LONG VIDEO GENERATION

As shown in Algorithm 1, at each timestep, the model performs denoising in a segment-wise manner. Let the audio embedding be $v_a^{[0,l]}$ of length $l$, the latent variable at timestep $t$ be $z_t^{[0,l]}$, and the segment length be $f$. We introduce a position-shift offset $\alpha$, where $\alpha < f < l$, determining how much the starting point shifts in each iteration. The accumulated shift offset is initialized as $\alpha_\beta = 0$. For each denoising step $t$ from $T$ down to 1, a new segment is selected with start $s = \alpha_\beta$, end $e = s + f$, and processed length $n = 0$. The algorithm iteratively processes each segment as long as $n < l$. For every segment, the pretrained AnyAvatar model predicts the next latent variable given the current latent and audio slice as:

$$z_{t-1}^{[s,e]} = \text{AAM}(z_t^{[s,e]}, v_a^{[s,e]}, t) \quad (4)$$

After each prediction, indices are updated as $s \leftarrow s + f$, $e \leftarrow e + f$, and $n \leftarrow n + f$. If these indices exceed the sequence length, i.e., if $s > l$ or $e > l$, we use circular padding: $s \leftarrow s \bmod l$ and $e \leftarrow e \bmod l$. After all segments are processed for a timestep, the accumulated offset is updated as $\alpha_\beta \leftarrow \alpha_\beta + \alpha$. In our experiments, we set the offset $\alpha$ to 5 for each timestep, and find that this setting effectively maintains coherence across segments. This segment-wise shifting strategy enables AnyAvatar to naturally bridge audio and visual contexts, resulting in continuous video generation that closely follows the audio prompts. The final denoised latent is $z_0^{[0,l]}$.

# 4 EXPERIMENT

## 4.1 EXPERIMENT SETTINGS

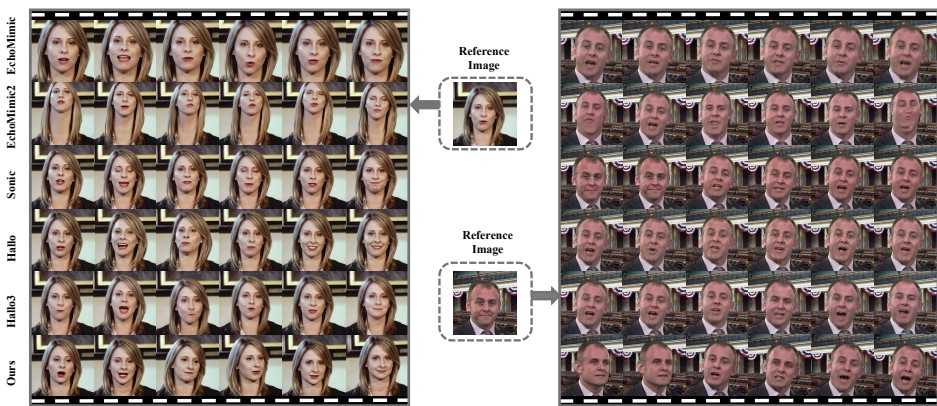

Figure 4: Qualitative comparison on the HTDF dataset.

**Implementation Details.** We use HunyuanVideo-I2V Kong et al. (2024) as the base model for AnyAvatar. The training process consists of two distinct stages. In the first stage, we train exclusively on audio-only data to establish fundamental audio-visual alignment. In the second stage, we implement a mixed training regime combining audio and image data in a 1:1.5 ratio to enhance motion stability. The resolution of the training data ranged from $704 \times 704$ to $704 \times 1216$. Throughout the training, we maintain fixed parameters for both LLaVA and 3D VAE while updating all other learnable parameters. We use 160 GPUs with 96GB of memory each, set the global batch size to 40, and the learning rate to 1e-5. More details are provided in the appendix A.5.

**Datasets.** To obtain high-quality training data, we use LatentSync Li et al. (2024) to filter out audio-visual asynchronous data and employ Koala-36M Wang et al. (2024b) to filter out data with low brightness or low aesthetics. Through our standardized data selection process, we obtain 500,000 samples with character audio, with a total duration of approximately 1,250 hours. During the testing stage, we select the publicly available portrait datasets CelebV-HQ Zhu et al. (2022) and HDTF Zhang et al. (2021) to evaluate the portrait animation capabilities of various methods. In addition, since there is currently no publicly available full-body animation test set, we construct our own full-body animation test set, which contains 250 videos covering 200 identities, involving different races, ages, genders, styles, and initial actions. More datasets details are provided in the appendix A.6.

**Evaluation Metrics and Compared Baselines.** We use the Q-align Wu et al. (2023) visual language model (VLM) to evaluate video quality (IQA) and aesthetic metrics (AES), and use FID Heusel et al. (2017) and FVD Unterthiner et al. to assess the distance between generated videos and real videos. In addition, we use the smoothness metric from VBench Huang et al. (2024) to evaluate video motion stability, employ Sync-C, Sync-D Chung & Zisserman (2017) to assess audio-visual synchronization and employ HKC, HKV Lin et al. (2024) are employed, to represent hand quality and motion richness respectively. Apart from objective metrics, we also conducted a subjective evaluation with 30 users. The thirty users rated the generated results across four dimensions: lip synchronization(LS), Identity Preservation (IP), Full-body Naturalness(FBN), and Facial Naturalness(FCN). To comprehensively assess the advancement of our method, we compared it with the current state-of-the-art audio-driven portrait animation methods, including Sonic, EchoMimic, EchoMimic-V2 and Hallo-3. For audio-driven full-body animation, we first compared Hallo3 Cui et al. (2024), FantasyTalking Wang et al.

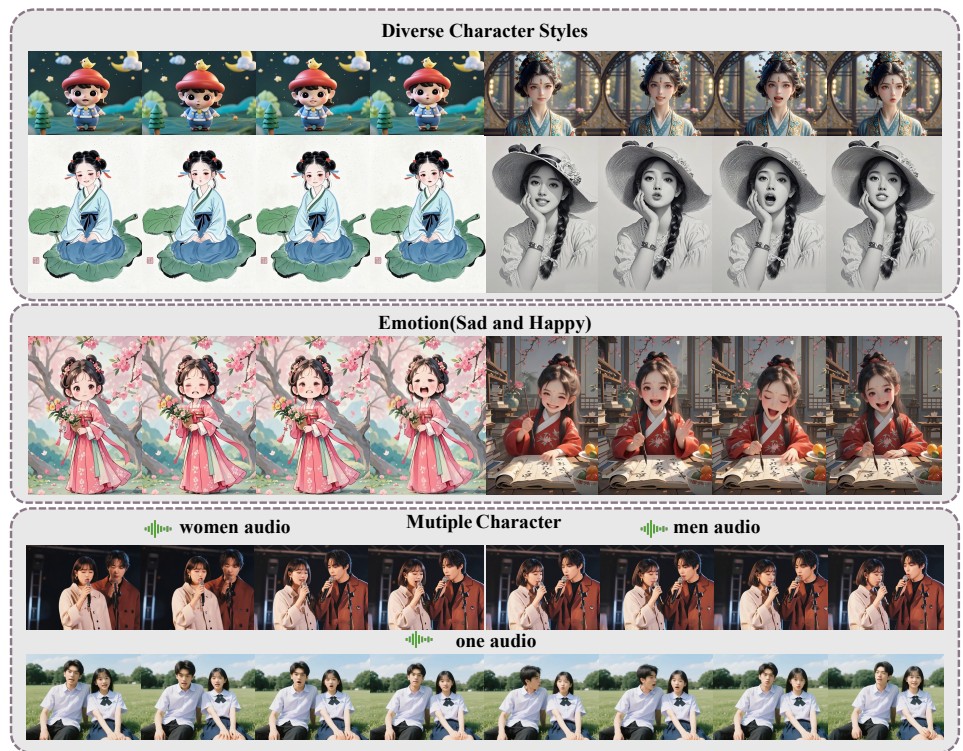

Figure 5: Visualization of videos generated by AnyAvatar on the wild dataset.

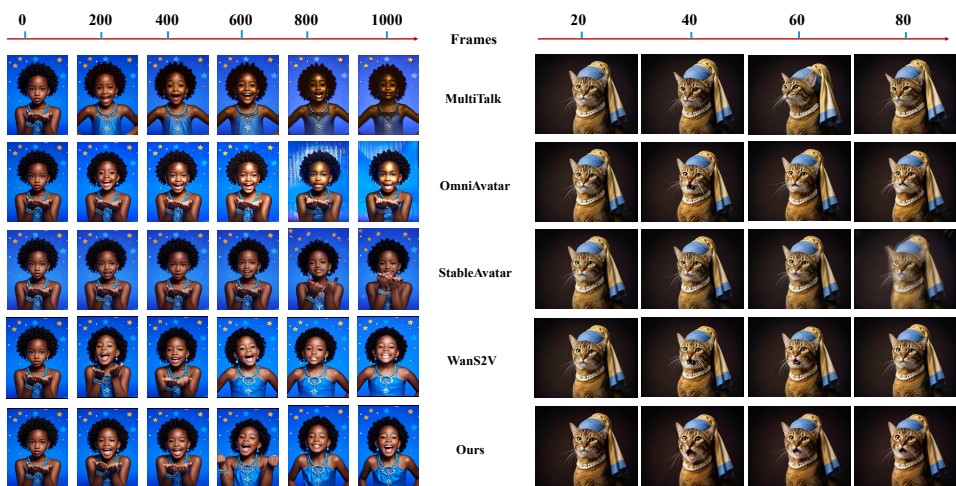

Figure 6: Qualitative comparison on the wild body dataset.

(2025b), Multitalk Kong et al. (2025), OmniAvatar Gan et al. (2025), StableAvatar Tu et al. (2025) and WanS2V Gao et al. (2025) on our proposed full-body animation test set.

## 4.2 COMPARISON WITH STATE-OF-THE-ART METHODS

**Qualitative Results.** We conducted qualitative comparisons with existing methods. For audio-driven portrait animation, we mainly compared our approach with Sonic, EchoMimic, EchoMimicV2, and Hallo-3 on the HDTF dataset, which primarily focuses on lip synchronization and facial expression accuracy. As shown in the figure 4, our method produces results with higher video quality, more natural and vivid facial expressions, and more aesthetically pleasing video effects on this dataset. For audio-driven full-body animation, The figure 5 demonstrates the effectiveness of our model across various styles of characters, emotion control, and audio-driven multi-character scenarios, showcasing

Table 1: **Quantitative comparisons with audio-driven portrait animation baselines.**

| Methods | IQA ↑ | AES↑ | Sync-C↑ | FID↓ | FVD↓ | Sync-D↓ |
|---|---|---|---|---|---|---|
| | | | CelebV-HQ / HDTF | | | |
| Sonic | 3.60 / 3.86 | 2.43 / 2.41 | 5.58 / **5.81** | 49.28 / 40.50 | **415.04** / 413.94 | **8.31** / 9.80 |
| EchoMimic | 3.39 / 3.64 | 2.25 / 2.23 | 3.41 / 4.07 | 46.74 / 45.38 | 450.98 / 410.05 | 9.99 / 10.31 |
| EchoMimic-V2 | 2.75 / 3.36 | 1.97 / 2.15 | 4.11 / 3.39 | 46.37 / 39.73 | 862.24 / 487.75 | 9.50 / 11.00 |
| Hallo-3 | 3.57 / 3.77 | 2.38 / 2.35 | 4.57 / 4.87 | 45.69 / 39.07 | 444.92 / 380.31 | 8.97 / 9.90 |
| **Ours** | **3.70 / 3.99** | **2.52 / 2.54** | **4.92** / 5.30 | **43.42 / 38.01** | 445.02 / **358.71** | 8.55 / **9.74** |

Table 2: **Quantitative comparisons with audio-driven full-body animation baselines.**

| Methods | IQA ↑ | AES↑ | Sync-C↑ | FID↓ | FVD↓ | Sync-D ↓ | HKC↑ | HKV↑ | FCN | FBN | IP | LS |
|---|---|---|---|---|---|---|---|---|---|---|---|---|
| Hallo3 | 4.345 | 2.771 | 5.131 | 50.122 | 629.943 | 9.942 | 0.623 | 0.268 | 2.91 | 2.59 | 4.28 | 3.61 |
| FantasyTalking | 4.631 | 3.023 | 3.682 | 58.243 | 677.672 | 11.213 | 0.771 | 0.379 | 3.43 | 3.49 | 4.65 | 4.21 |
| OmniHuman-1 | 4.652 | 2.995 | 5.343 | 49.681 | 719.401 | 9.774 | 0.839 | 0.310 | **4.11** | 4.18 | 4.79 | 4.61 |
| OmniAvatar | 4.607 | 3.004 | 7.121 | 54.675 | 654.104 | 7.987 | 0.754 | 0.223 | 3.65 | 3.12 | 4.32 | 4.22 |
| StableAvatar | 4.638 | 3.015 | 7.234 | 55.644 | 639.006 | 8.690 | 0.794 | 0.386 | 3.77 | 3.61 | 4.79 | 4.59 |
| Multitalk | 4.361 | 2.864 | 6.987 | 52.330 | **613.213** | 8.637 | 0.785 | 0.388 | 3.82 | 3.56 | 4.75 | 4.55 |
| WanS2V | **4.812** | 3.007 | 6.989 | 49.777 | 921.998 | **7.957** | 0.783 | **0.413** | 4.02 | **4.50** | 4.80 | 4.59 |
| **Ours** | 4.668 | **3.036** | **7.534** | **49.380** | 650.541 | 8.535 | **0.849** | 0.390 | 3.91 | 3.88 | **4.84** | **4.65** |

Table 3: Ablation on CIIM.

| Methods | VQ ↑ | MD ↑ | IP ↑ | LS ↑ |
|---|---|---|---|---|
| Token | 2.863 | 3.585 | 4.402 | 4.239 |
| Token + Channel | 4.412 | 2.336 | **4.576** | **4.431** |
| Token + Add | **4.486** | **4.127** | 4.289 | 4.161 |

Table 4: Ablation on the Methods to Inject Face Masks.

| Methods | BD↑ | SB↑ | DB↑ | IQA ↑ | AES↑ | FID↓ | FVD↓ |
|---|---|---|---|---|---|---|---|
| w/o mask | 0.0027 | 100% | 87% | 4.803 | 3.587 | **72.124** | 1205.123 |
| w token | 0.0025 | 100% | 85% | 4.764 | 3.335 | 73.124 | 1400.153 |
| w mask | **0.0028** | 100% | **90%** | **4.815** | **3.589** | 74.087 | **1202.491** |

the validity of our approach. Then we mainly compared our method with other methods on the wild full-body dataset. As shown in Figure 6, the videos generated by our method demonstrate more natural variations in the foreground, background, and character movements, while also achieving more accurate lip synchronization and better character consistency, resulting in higher overall video quality. These improvements are attributed to the focused design of the audio adapter module and the introduction of the character image injection module. Notably, we tested 50-second videos to compare the long video generation capabilities of different methods. It is evident that our method exhibits stronger stability in 50-second video generation. Additionally, other methods often fail to animate non-human characters. Therefore, our approach is better suited to meet the demands of practical application scenarios. More comparative and visual results are provided in the appendix.

**Quantitative Results.** To thoroughly validate the superiority of our method in audio-driven portrait animation, we compared our approach with baseline methods on various evaluation metrics using the CelebV-HQ and HDTF test sets. As shown in the Table 1, the results demonstrate that our method achieves the best performance, proving the effectiveness of our approach in audio-driven portrait animation and showcasing its capability in audio synchronization. Meanwhile, to verify the superiority of our method in audio-driven full-body animation, we conducted a comparison with baseline methods on various evaluation metrics using our proposed test set. As shown in Table 2, the experimental results demonstrate that our method achieves the best performance on most evaluation metrics, proving its effectiveness in audio-driven portrait animation generation and showcasing its capability in audio-visual synchronization.

**User Study.** To further validate the effectiveness of our proposed method, we conducted a subjective evaluation on the wild full-body animation dataset. Each participant assessed four key dimensions: lip synchronization (LS), identity preservation (IP), full-body naturalness (FBN), and facial naturalness (FCN). A total of 30 participants rated each aspect on a scale from 1 to 5. For each method, we

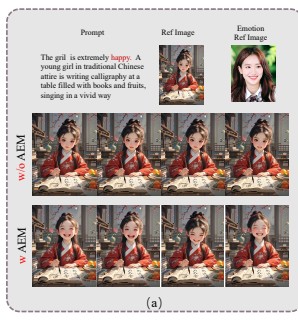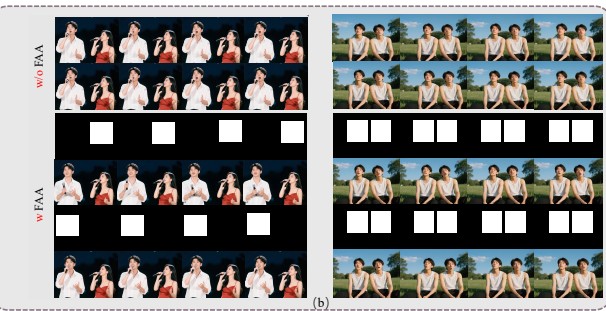

Figure 7: (a) Ablation on Audio Emotion Module. (b)Ablation on Face-Aware Audio Adapter.

generated 30 videos, and each participant was required to evaluate videos from all methods. As shown in Table 2, the results indicate that AnyAvatar outperforms existing baseline methods in the IP and LS evaluation dimensions. Since OmniHuman-1 is not open source and its online service includes super-resolution operations, there is a natural visual advantage in subjective evaluations. In addition, our method inherits some inherent issues of Hunyuanvideo. Therefore, in terms of FCN and FBN, our scores show certain deficiencies compared to OmniHuman-1.

**Ablation on Character Image Injection Module.** We subjectively evaluated three Character Image Injection Modules across four dimensions: Lip Synchronization (LS), Video Quality (VQ), Identity Preservation (IP), and Motion Diversity (MD). As shown in Table 3: (1) Token concat excels at enhancing video dynamics, but performs poorly in terms of IP and VQ. (2) Token concat + channel concat method ensures the consistency of characters, backgrounds, and foregrounds, but it can be seen from the MD that the dynamic motion is greatly restricted. (3) Token concat + add maintains video dynamic motion while also preserving the consistency of characters, backgrounds, and foregrounds.

**Ablation on Audio Emotion Module.** Figure 7**(a)** demonstrates that when only text guidance is used, the model cannot generate facial expressions, while the introduction of the AEM enables better alignment between the emotions conveyed by the audio and the character's facial expressions.

**Ablation on Face-Awared Audio Adapter.** The figure 7**(b)** shows that without using a mask, the characters are driven by the audio randomly; whereas after applying the mask, only the masked character is affected by the audio, enabling precise audio control for multiple characters.

**Ablation on the Methods to Inject Face Masks.** As shown in the Table 4, we compared the method of injecting masks into Video Tokens, commonly used in inpainting task. To demonstrate that introducing masks does not degrade the video backgrounds, we additionally introduced three metrics. We selected 50 images with clean backgrounds to evaluate the success rate of each method in preserving static backgrounds (SB), and 50 images with complex backgrounds to assess their performance in maintaining physical consistency (DB). We also measure the optical flow score of the background to evaluate background dynamics (BD). Overall, our method outperforms the other two approaches in both background evaluation metrics and overall video quality. Moreover, the results of BD, SB, and DB clearly indicate that the introduction of masks does not lead to foreground or background distortion. Furthermore, the IQA, AES, FID, and FVD demonstrate that our method achieves higher overall consistency and better performance compared to the approach without masks.

## 5 CONCLUSION

In this paper, we propose AnyAvatar, an audio-driven human animation method that achieves both high character consistency and dynamic motion. We introduce a character image injection module resolves the inherent trade-off between dynamism and consistency by adaptively balancing these objectives, significantly enhancing the naturalness and diversity of generated videos. To ensure alignment between the audio's emotional tone and character expressions, we introduce the Audio Emotion Module which transfers affective cues from emotion reference images to the target animation. For multi-character scenarios, our method employs latent-space masking to localize audio-driven animation to specific face regions, enabling independent control of different characters through targeted mask modulation. Extensive qualitative and quantitative experiments demonstrate that AnyAvatar outperforms existing methods in video dynamism, subject consistency, lip-sync accuracy, audio-emotion-expression alignment, and multi-character scenarios.

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

# A APPENDIX

In this appendix, we offer further details on preliminary, implementation details, present additional experimental results, and limitations and societal impacts, structured as follows:

- The Use of Large Language Models (Sec. A.1);
- Reproducibility statement (Sec. A.2);
- Ethics statement (Sec. A.3)
- Preliminary (Sec. A.4);
- Implementation Details (Sec. A.5);
- Datasets Details (Sec. A.6);
- More Visualization Results (Sec. A.7).
- Limitations (Sec. A.8)
- Societal Impacts (Sec. A.9)

## A.1 THE USE OF LARGE LANGUAGE MODELS

We acknowledge the use of a large language model (LLM) to assist in the preparation of this manuscript. The LLM's role was strictly limited to improving grammar and refining language. It did not contribute to any of the core research components, such as the initial ideas, experimental design, data analysis, or interpretation of the results.

## A.2 REPRODUCIBILITY STATEMENT

We have explained the implementation of AnyAvatar in detail in Sec. 4.1 and Sec. A.5. The code and dataset pipeline used in this work will be open-source online.

## A.3 ETHICS STATEMENT

**Source of Test Data.** In the visualization results presented in the main text and supplementary materials, some of the character images and driving audio are sourced from open-source test sets, while the rest are generated by AI. Before collecting users' image data, we require users to sign an informed consent form, explicitly authorizing the use of their image data for academic research and non-commercial secondary dissemination. This fundamentally avoids infringement of others' portrait rights and copyrights.

**User study.** All user study experiments are conducted in strict accordance with national laws, and reasonable compensation is provided to participants.

**Training Data Standards.**

- **Data Collection.** When collecting the data, we were careful to only include videos that—to the best of our knowledge, were intended for free use and redistribution by their respective authors. That said, we are committed to protecting the privacy of individuals who do not wish their videos to be included.
- **Dataset Balancing.** To eliminate various biases in the dataset, we manually increased the diversity and representativeness of the data, for example, by adding more data for minority groups/races and reducing the amount for majority groups/races. This helps reduce risks such as reinforcing stereotypes, generating offensive content, or producing lower-quality results for specific groups.
- **Data Screening and Filtering.** We removed harmful content from the collected training dataset and introduced prompt and video filters to prevent users from generating inappropriate content. For training data selection, we first use open-source tools for preliminary screening, and then employ ten staff members to carry out a detailed manual review over a period of half a month. During this process, we negotiate reasonable compensation standards with the staff in advance, and final payment is settled based on each individual's actual workload.

**Model Anti-Abuse Measures.** To prevent misuse of the model by users, we have implemented and will continue to improve multiple preventive measures, including:

- **User Behavior Guidelines.** To prevent harassment and bullying of others, we require users not to upload photos of others without their consent. Content Identification: All content generated by AnyAvatar will be clearly marked as synthetic (e.g., via watermarking) to prevent it from being used to mislead others.

- **Safety Assessment and Monitoring.** We will continue to expand our safety assessment tools, conduct regular risk analyses, and strictly review and retrospectively evaluate new use cases. At the same time, we will develop a monitoring system that combines automation and human oversight to promptly detect and prevent potential abuse.

- **Compliance and Distribution.** We will impose strict usage requirements on the model to ensure it is only used for legal and compliant scientific research purposes. We will continue to conduct relevant research, actively listen to community feedback, dynamically improve our ethical guidelines, and do our utmost to eliminate potential negative social impacts, promoting the healthy and responsible development of technology.

**Regarding culture bias.** There are significant differences in emotional expression across cultures, yet most current emotion recognition and generation models remain centered on Western norms, which can introduce cultural bias. Most of these technologies perform inconsistently across different datasets and are highly sensitive to label definitions, reflecting inherent biases in the training data. Although the AnyAvatar model does not directly classify emotions, its training data and generation mechanisms may still inherit such biases. In the future, we plan to incorporate multicultural datasets and culturally adaptive algorithms to improve the model's fairness and applicability across diverse backgrounds.

## A.4 PRELIMINARY

**Diffusion Transformer.** The Diffusion Transformer (DiT) is a diffusion model designed based on the Transformer architecture. With the emergence of SoRA Liu et al. (2024), we have seen its tremendous potential in the field of video generation. Multi-Modal DiT (MM-DiT) is an improved version of the DiT structure, and the main difference between MM-DiT and DiT lies in the way conditions are injected. DiT uses a cross-attention mechanism for text injection, while MM-DiT connects textual information with images or videos to perform joint attention. Specifically, we adopt HunyuanVideo as our backbone. This model uses a causal 3DVAE to compress videos in both temporal and spatial dimensions, and employs LLaVA to encode textual information and obtain text embeddings. The textual and video information are then jointly input into MMDiT.

**Identity Enhancement and Motion Dynamics.** Firstly, we resize the target image to match the dimensions of the video frames. We then use the pretrained 3DVAE from HunyuanVideo to map the reference image $R$ from image space to the latent space, obtaining the reference image latent $v_R \in \mathbb{R}^{w \times h \times c}$, where $w$ and $h$ denote the width and height of the latent, and $c$ is the feature dimension. Similarly, we encode the noise video using the 3D VAE to obtain the video latent $v_{\text{noise}} \in \mathbb{R}^{f \times w \times h \times c}$ where is the the number of video frames. Next, we process $v_R$ with Tokenizer2 $K_2$ to obtain $t_R \in \mathbb{R}^{wh \times c}$ and $t_{\text{noise}} \in \mathbb{R}^{fwh \times c}$, respectively. We then replicate the reference image $T$ times (where $T$ is the original video length) to obtain $i_r$, and use the 3DVAE together with Tokenizer1 $K_1$ (initialized with the weights of Tokenizer2) to obtain $t_r \in \mathbb{R}^{fwh \times c}$. We add $t_r$ and $t_{\text{noise}}$ element-wise, and concatenate the result with $t_R$ along the token dimension to form the final input $p$, as shown below:

$$p = \text{TokenCat}\left(\{K_1(t_r) + K_2(t_{\text{noise}})\}, t_R\right) \tag{5}$$

Thanks to the strong temporal modeling prior of HunyuanVideo, identity information can be efficiently propagated along the time axis. Therefore, we assign 3D-RoPE (Su et al., 2024) positional encoding to the concatenated image latent. In the original HunyuanVideo, video latents are assigned 3D-RoPE along the time, width, and height axes; for a pixel at position $(f, i, j)$ (where $f$ is the frame index, $i$ is the width, and $j$ is the height), the RoPE is $RoPE(f, i, j)$. For the image latent, to enable effective broadcasting of identity information along the temporal sequence, we place it at the $-1$-th frame,

i.e., before the first frame with time index 0. Furthermore, inspired by Omnicontrol (Tan et al., 2024) in controllable image generation, to prevent the model from simply copying and pasting the target image into the generated frames, we introduce a spatial shift for the image latent, as follows:

$$RoPE_{z_I}(f, i, j) = RoPE(-1, i + w, j + h). \tag{6}$$

The LLaVA model, as a multi-modal understanding framework, is designed to capture the correlation between text and image, primarily extracting high-level semantic information such as category, color, and shape, while often overlooking finer details like text and texture. However, in video customization, identity is significantly determined by these image details, making the LLaVA branch alone insufficient for identity preservation. To address this, we propose an identity enhancement module. By concatenating video latents with the target image over the time axis, and leveraging the video model's efficient information transmission capability in the temporal dimension, we can effectively enhance video identity consistency.

**Training.** During the training process, we employ the Flow Matching Lipman et al. (2022) framework to optimize our video generation model. Specifically, we first extract the latent representation of the video, denoted as $v_i$, along with its corresponding reference image $R$. To introduce stochasticity, we sample a time step $t \in [0, 1]$ from a logit-normal distribution Esser et al. (2024). We then initialize the noise vector $z_0 \sim \mathcal{N}(0, I)$ from a standard Gaussian distribution. The training sample at time $t$, $v_t$, is constructed by linearly interpolating between the initial noise $z_0$ and the target latent $v_i$.

The model is trained to predict the velocity $u_t = \frac{dz_t}{dt}$ at each time step, conditioned on the reference image $R$. This velocity guides the sample $v_t$ towards the target latent $v_0$. During optimization, the model outputs a predicted velocity $\lambda_t$, and the parameters are updated by minimizing the mean squared error between $\lambda_t$ and the ground-truth velocity $u_t$. The overall generation loss is defined as:

$$\mathcal{L}_{\text{generation}} = \mathbb{E}_{t, x_0, x_1} \left\| \lambda_t - u_t \right\|^2. \tag{7}$$

This training strategy enables the model to effectively learn the underlying data distribution and generate high-quality, customized video content conditioned on the reference image.

## A.5 IMPLEMENTATION DETAILS

**Long Video Generation.** The HunyuanVideo13B model Kong et al. (2024) can only generate videos with 129 frames, which is often shorter than the audio length. To tackle the challenge of generating long videos, we use the Time-aware Position Shift Fusion method from Sonic Ji et al. (2024). We successfully adapt this method to the HunyuanVideo13B model, which is based on the MM-DiT architecture, and achieve good results. This fusion strategy is simple yet effective, as it does not add any extra inference or training costs. It helps to reduce issues like jitter and abrupt transitions during video generation.

As shown in Algorithm 1, at each timestep, the model takes a segment of the audio as input to predict the corresponding latent. It uses a starting offset to smoothly connect with the segment from the previous timestep, shifting forward by $\alpha$ steps each time. We set the offset $\alpha$ to 5 at each timestep, and our experiments show that this is an effective choice. This approach allows AnyAvatar to naturally bridge the context, enabling continuous video generation that follows the audio prompts.

**Drive Multi-Character Talking.** AnyAvatar can achieve lip synchronization for multiple characters (such as ID A and ID B) with single forward pass. Specifically, we input an audio clip into the model, where the first half contains the audio of ID A and the second half contains the audio of ID B. We simply apply the mask for the first half to the facial region of ID A, and the mask for the second half to the facial region of ID B. In this way, multiple characters can be driven in a single inference. Our method enables driving multiple people to speak simultaneously using a single mask, but they can only speak the same lines according to the audio. In other words, if there are two people within one mask, they can speak the same sentence at the same time. However, it is currently not possible to support scenarios where different characters speak different lines simultaneously or where interruptions occur during speech. We believe this is a highly meaningful research direction, and we will focus on this area in the future to further explore more possibilities in more natural, multi-speaker scenarios.

---

**Algorithm 1** Long Video Fusion

---

**Require:** Audio embedding $v_a^{[0,l]}$ with length $l$, denoising steps $T$, initial noisy latent $z_T^{[0,l]}$, pretrained AnyAvatar model $AAM(\cdot)$ for sequence length $f$, position-shift offset $\alpha < f < l$.
**Ensure:** Denoised latent $z_0^{[0,l]}$.
  1: Initialize accumulated shift offset $\alpha_\beta = 0$.
  2: **for** $t = T, \cdots, 1$ **do**
  3:                                                          // Denoising loop
  4:     Initialize start point $s = \alpha_\beta$, end $e = s + f$, processed
        length $n = 0$.            // Start from new position for each timestep.
  5:     **while** $n < l$ **do**
  6:                                                  // Sequence loop
  7:        $z_{t-1}^{[s,e]} = AAM(z_t^{[s,e]}, v_a^{[s,e]}, t)$
  8:        $s \leftarrow s + f, e \leftarrow e + f, n \leftarrow n + f$.       // Move to next clip non-overlapping
  9:        **if** $s > l$ **or** $e > l$ **then**
10:           $s \leftarrow s\%l, e \leftarrow e\%l$.           // Padding circularly
11:        **end if**
12:     **end while**
13:     $\alpha_\beta \leftarrow \alpha_\beta + \alpha$.              // Accumulate shift offset
14: **end for**
15: **return** Denoised latent $z_0^{[0,l]}$.

---

### A.6 DATASETS DETAILS

**Data sources:** Open-source data (OpenhumanVid Li et al. (2025)) and self-collected data. When collecting the data, we were careful to only include videos that to the best of our knowledge were intended for free use and redistribution by their respective authors. That said, we are committed to protecting the privacy of individuals who do not wish their videos to be included.

**Data distribution:** Our data is divided into video data with audio and video data without audio. Among the video data with audio, there are **50,000** samples of two-person dialogues, **190,000** samples of single-person speech, and **10,000** samples of animation. For the video data without audio, we do not distinguish between categories, and there are a total of **250,000** samples.

### A.7 MORE VISUALIZATION RESULTS

Figure 8 shows the results of our method in multiple characters scenarios such as crosstalk, singing, and walking conversations, demonstrating the robustness of our model.

Figure 9 presents visualizations of realistic human images. From this scene, it can be seen that our model is able to maintain good character consistency while enhancing dynamics, further demonstrating the effectiveness of our character image injection module.

Figure 10 showcases the generation results of our method applied to characters with diverse styles. The results show that our method generalizes well across various styles, including LEGO, chinese painting, anime, and pencil sketch.

Figure 11 demonstrates the precise control of emotions achieved by our method. It can be seen that our model has a good understanding of emotions such as happiness, sadness, excitement, and anger. This enables us to generate human animation videos that better align with the emotions conveyed by the audio, further demonstrating the unique capabilities of our model compared to previous audio-driven human animation methods.

In summary, compared to previous audio-driven human animation methods Wang et al. (2025b); Lin et al. (2025a); Jiang et al. (2024); Cui et al. (2024); Ji et al. (2024), our approach offers more practical features such as multi-character and emotion control audio-driven human animation. At the same time, it also outperforms previous methods in terms of character consistency and video dynamics. These advancements highlight the state-of-the-art performance and innovative design of our model.

A.8 LIMITATIONS

Firstly, our current approach relies on emotion reference images to drive the character's emotions, rather than allowing the model to infer and generate emotions directly from the audio. This leads to two main issues: (1) increased complexity for users during operation, and (2) the inability to reflect dynamic emotional changes within the video. Since each reference image corresponds to only one emotion, multiple emotions in a single audio segment may result in generation errors. Therefore, exploring methods to directly extract emotions from audio and generate corresponding emotional character videos is a promising direction for future research.

Secondly, we currently use HunyuanVideo13B Kong et al. (2024) as our base model, while FantasyTalking Wang et al. (2025b) employs Wan14B Wang et al. (2025a). Regardless of the base model, the inference process is time-consuming. For instance, generating a 10s video at 720×1216 resolution (with 50 inference steps) takes approximately 60 minutes, which is far from meeting the requirements of real-time applications. Thus, improving the model's generation speed to achieve real-time performance is one of our key future objectives. This will facilitate the application of our model in scenarios with higher real-time demands, such as live streaming and interactive real-time applications.

Finally, exploring interactive human animation capable of real-time feedback is a promising research direction. This is expected to further expand the application of our method for users. This direction requires our model not only to possess strong content generation capabilities but also to have a solid understanding and contextual awareness, enabling fast and contextually appropriate responses to users. Our current focus is on developing an offline, high-precision, audio-driven model that achieves state-of-the-art performance. Of course, there are already several mainstream acceleration schemes for DiT architectures in the industry, and we have experimented with some of these methods. By combining these with hardware acceleration, we are optimistic about achieving real-time video generation with our model. For example, we have explored various training-free acceleration methods (such as Jenga), and achieved a 4x speedup on our model (for instance, with a reference image at 704×1216 resolution, generating a 5-second video with 30 inference steps originally took 60 minutes, but now only takes 15 minutes after acceleration). Although acceleration is not the main focus of our current work, we will continue to explore efficient acceleration techniques for large models, striving to achieve real-time applications as soon as possible.

A.9 SOCIETAL IMPACTS

Real-time interactive digital humans Ao (2024) have become a major focus in the fields of artificial intelligence. However, their development has not yet reached its full potential due to several technical limitations. On one hand, current generative models still struggle to produce diverse and natural actions and expressions, making it difficult to achieve truly lifelike interactions. On the other hand, many high-performance models are extremely large in terms of parameter count, resulting in slow inference speeds that cannot meet the demands of real-time generation. These challenges significantly hinder the practical deployment of interactive digital humans.

Against this backdrop, audio-driven human animation technology plays a crucial role in advancing the development of interactive digital humans. Some technologies, such as AnyAvatar, provides strong support for improving the quality of real-time digital human performance. By leveraging it, these systems can significantly enhance the naturalness of digital human conversations and effectively address issues such as unnatural facial expressions and movements. The application of such technologies greatly improves the expressiveness and emotional richness of digital humans, laying a solid technical foundation for the further growth of the industry. We believe that, with ongoing technological advancements, audio-driven human animation will be widely adopted in various real-world scenarios, enabling digital humans to deliver more intelligent and realistic interactive experiences.

Impacts on Employment: This research enables the generation of high-quality audio-visual synchronized videos, which can greatly improve the efficiency of film and television professionals and reduce repetitive labor, thereby shortening working hours and significantly promoting the development of the video production industry. While it may lead to a reduction in certain job positions, overall, this research plays a substantial role in advancing the industry.

Impacts on Public Property and Democratic Institutions: We have already considered these aspects and have taken various measures to prevent harm to public property and democratic institutions. We add watermarks and content tracing to the generated videos, and introduce prompt and video filters to prevent users from generating inappropriate content that could mislead others. At the same time, we have implemented a public figure review mechanism, prohibiting users from uploading images of public figures to ensure the authenticity of political leaders and topics.

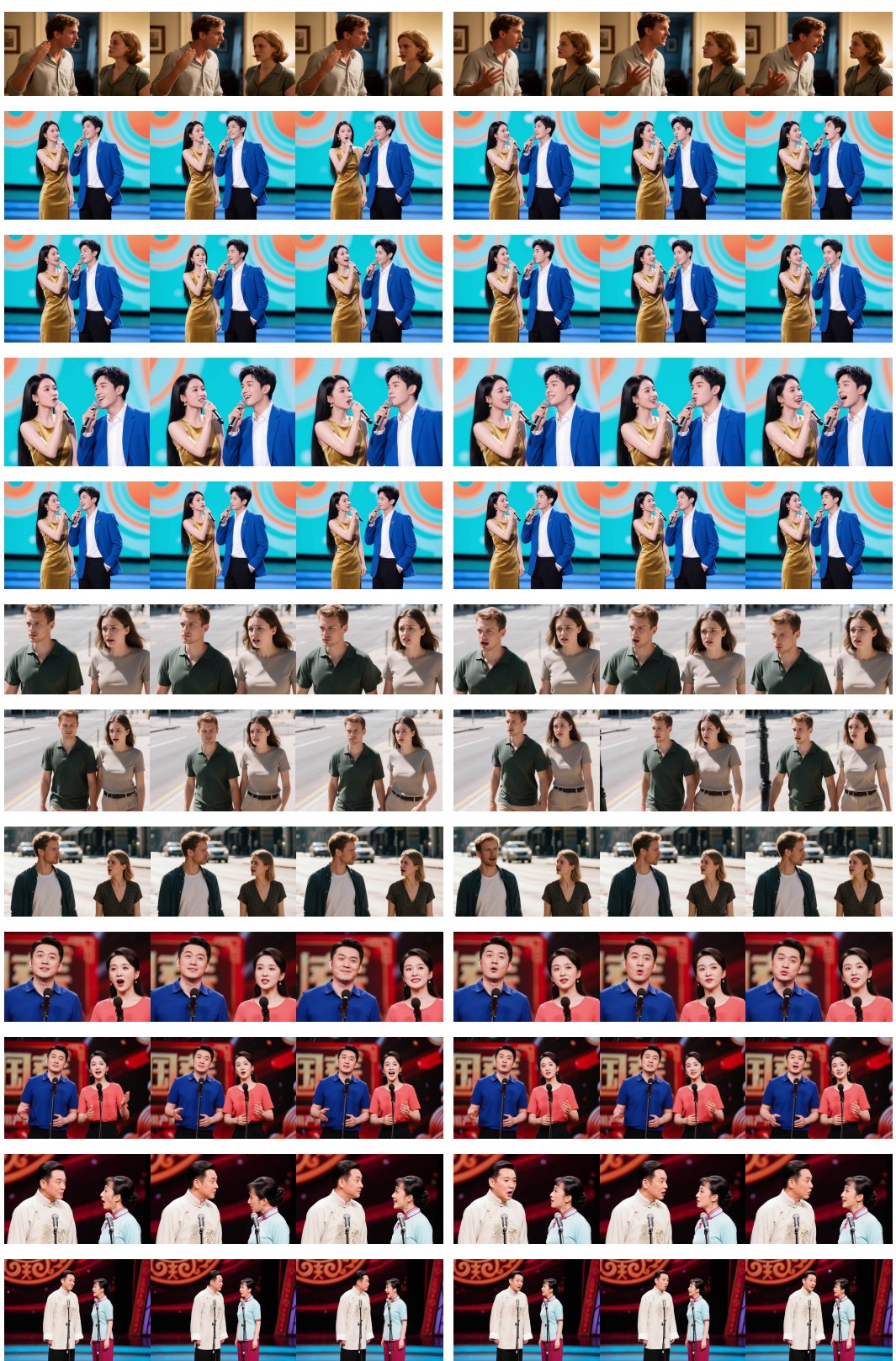

Figure 8: More visualizations on multi-character audio-driven human animation.

**Ref Image**                    **Output Video**

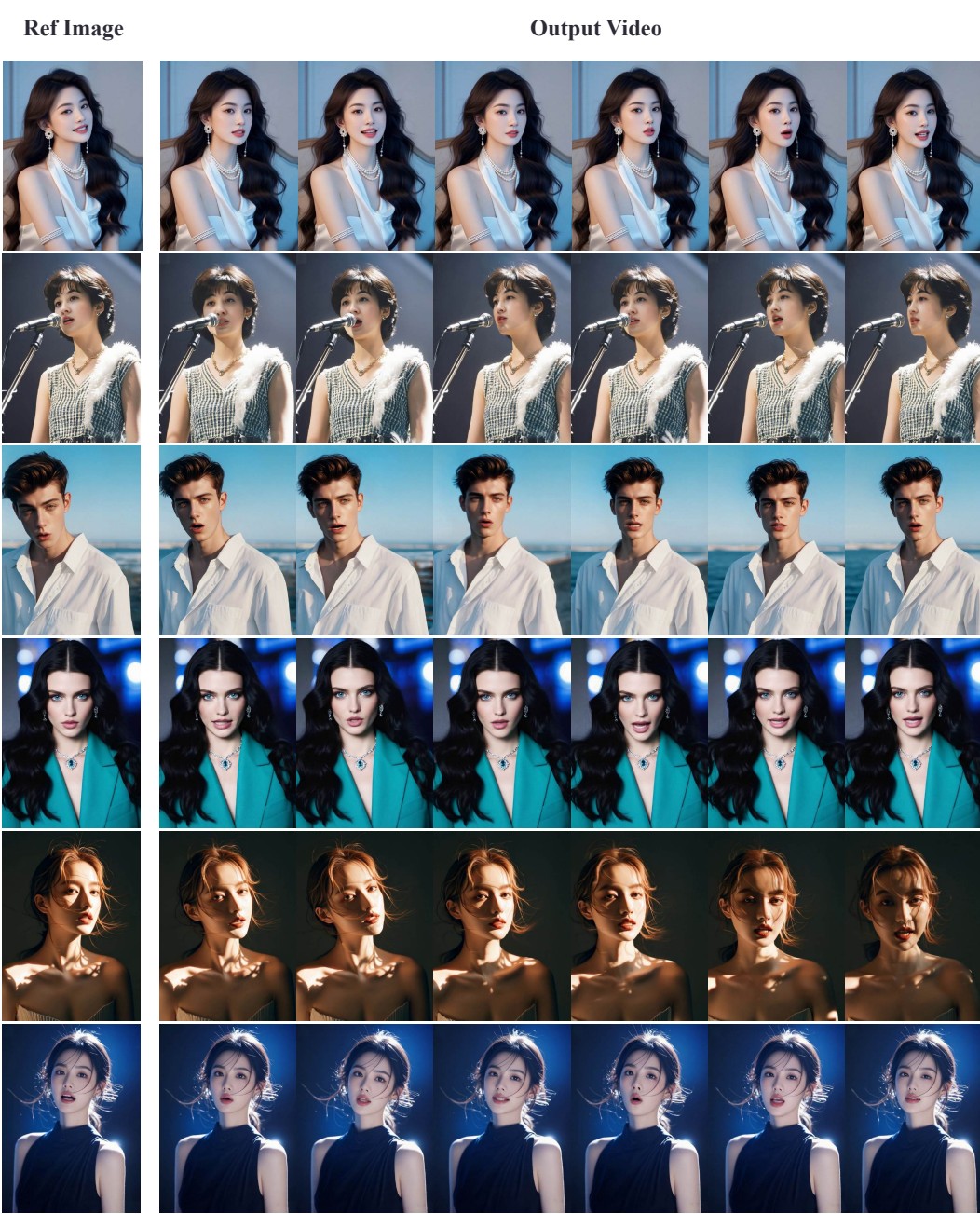

Figure 9: More visualizations on realistic scenarios.

**Ref Image**                                    **Output Video**

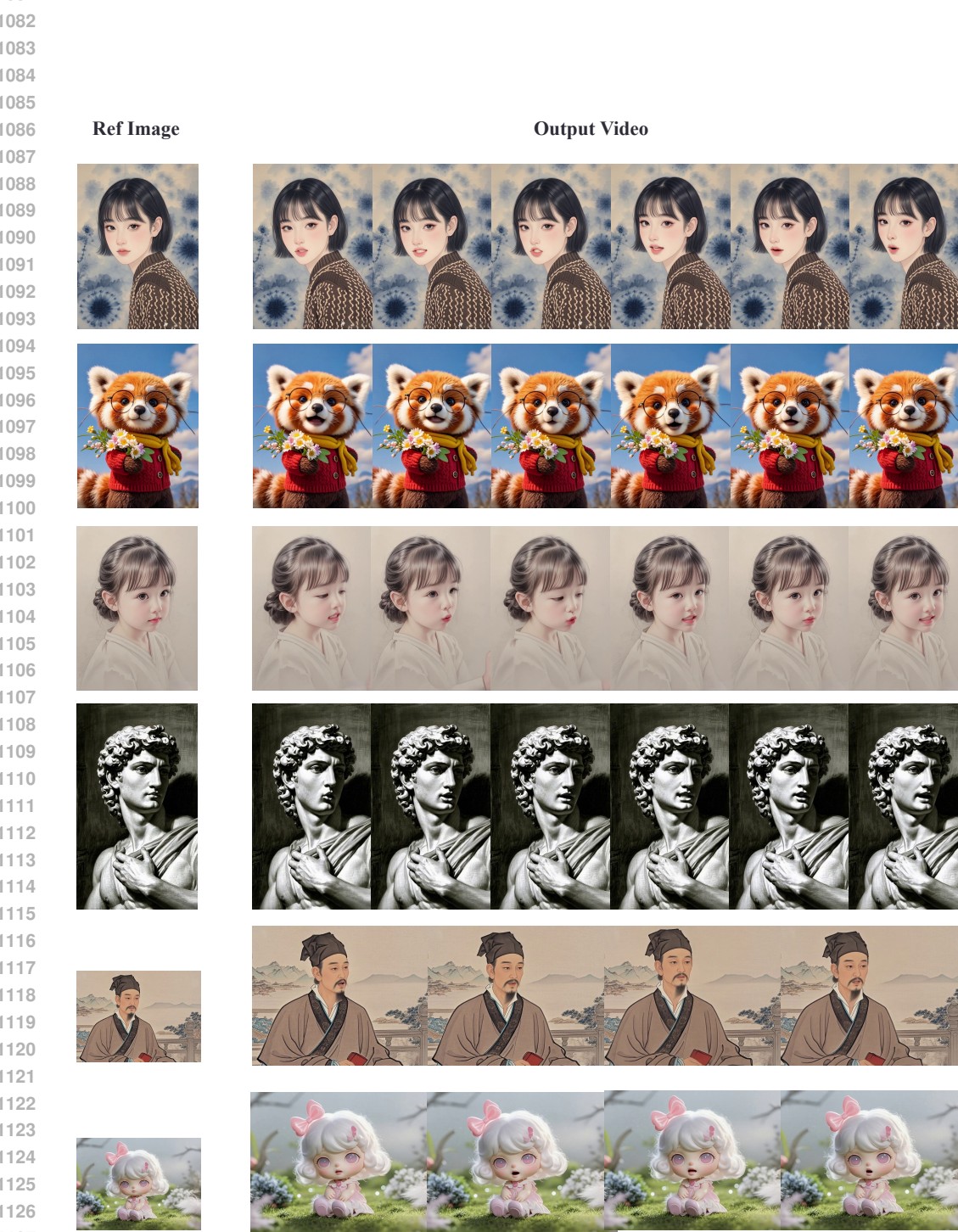

Figure 10: More visualizations on diverse character styles

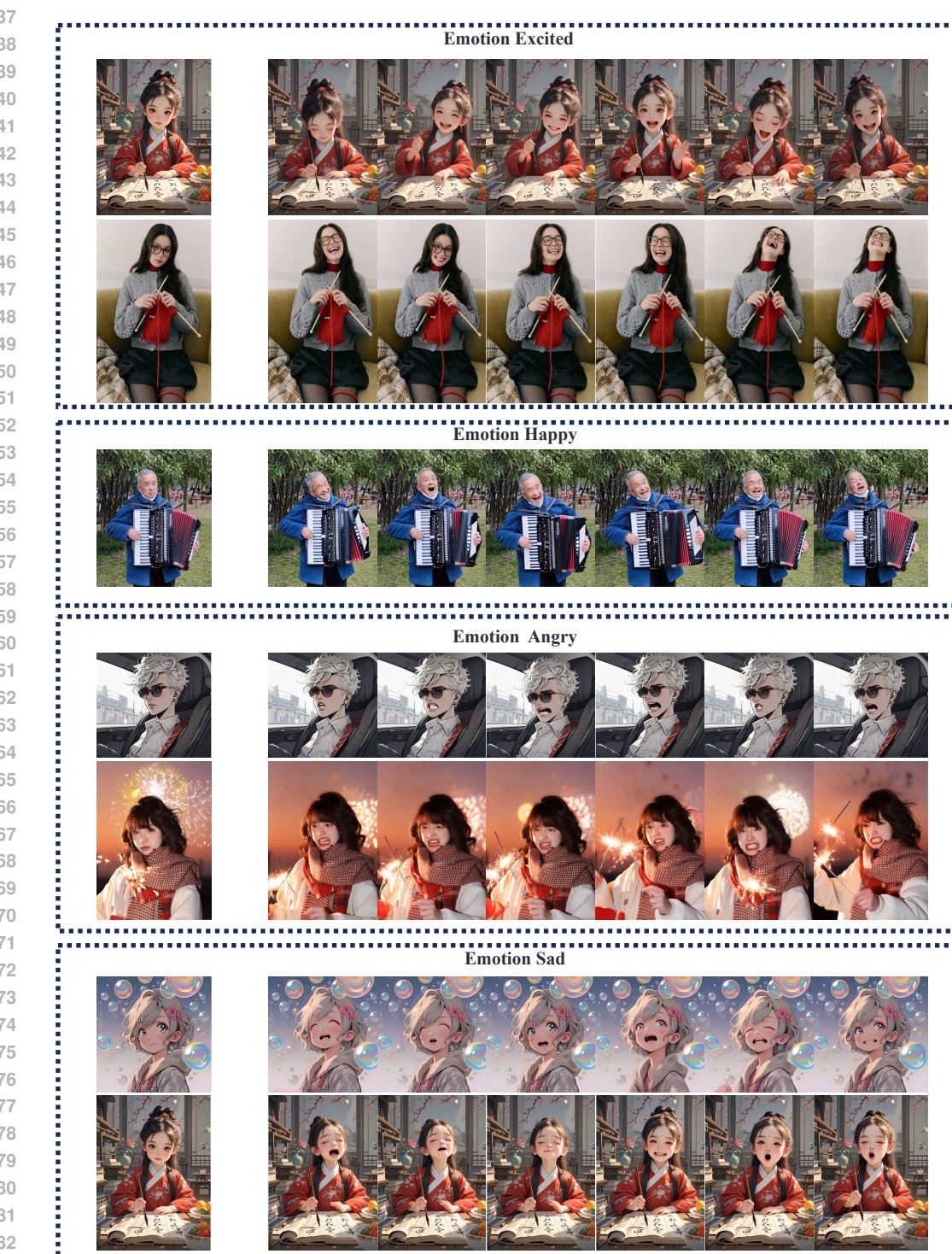

Figure 11: More visualizations on emotion control.

