# OpenReview forum: "AnyAvatar: Dynamic and Consistent Audio-Driven Human Animation for Multiple Characters"
_ICLR.cc/2026/Conference — ICLR 2026 Conference Withdrawn Submission_

### Official Review · Reviewer_xbfC · 2025-10-16

**Soundness:** 2
**Presentation:** 3
**Contribution:** 3
**Rating:** 6
**Confidence:** 5

**Summary:**

The paper introduces AnyAvatar, a new method for audio-driven human animation built upon a multimodal diffusion transformer (MM-DiT) architecture. The goal is to generate dynamic, emotionally expressive, and consistent videos for single or multiple characters from just a character image and an audio track. The authors identify three key challenges in the field: balancing dynamic motion with character consistency, ensuring accurate emotion alignment between audio and video, and enabling multi-character animation.

**Strengths:**

- Tackles a Key Underexplored Problem: The paper focuses on multi-character audio-driven animation, which is a significant and challenging area that many contemporary methods do not address. The Face-Aware Audio Adapter is an interesting approach to this problem.

- Strong Qualitative Results: The visual examples provided in the paper are compelling. The model demonstrates the ability to generate high-quality, coherent videos across a wide variety of character styles (realistic, anime, sketch), emotions, and multi-character scenarios.

- Well-Structured and Easy to Follow: The paper is clearly written and easy to understand. The authors effectively break down the problem and their proposed solutions into distinct, logical modules.

- Addresses Multiple Challenges Simultaneously: The work is ambitious in scope, attempting to solve challenges related to motion dynamics, identity preservation, emotion, and multi-character generation within a single unified framework.

**Weaknesses:**

- Clarity and Rationale of the Emotion Module (AEM): The logic behind the AEM is unclear and seems contradictory. It aims to align video emotions with the audio but relies on a static emotion reference image. This approach isn't well-justified in the introduction or methods sections. It raises questions about why this is preferable to extracting emotional cues directly from the audio, and it inherently limits the animation to a single, static emotion throughout the clip.

- Weak Ablation Study for AEM: The ablation for the AEM in Figure 7(a) only compares its use against text-only guidance. A proper ablation would compare it to other methods of injecting emotion (e.g., directly from audio features) to prove the superiority of using a reference image.

- Incomplete Literature Review: The related work section is missing several relevant papers, including SadTalker [1], AniPortrait [2], and Keyface [3]. The omission of Keyface is particularly notable, as it proposed a method for conditioning emotion based on continuous valence and arousal values, which is a more nuanced approach than the one presented here.

- Lack of Detail in Results Analysis: The analysis of the quantitative results in Tables 1 and 2 is superficial. The text primarily states that the method is superior without providing insights into why it performs better on certain metrics or discussing the cases where it is not state-of-the-art.

- Insufficient Explanation of the Face Masking Mechanism: The practical implications of the Face-Aware Audio Adapter are not discussed. The ablation in Figure 7(b) is low-resolution and doesn't clearly demonstrate its benefits or potential failure cases. The paper does not address whether face masks are needed at inference time and what the pros and cons of such a dependency are.

[1] Zhang, W., Cun, X., Wang, X., Zhang, Y., Shen, X., Guo, Y., Shan, Y. and Wang, F., 2023. Sadtalker: Learning realistic 3d motion coefficients for stylized audio-driven single-image talking face animation. In Proceedings of the IEEE/CVF conference on computer vision and pattern recognition (pp. 8652-8661).

[2] Wei, H., Yang, Z. and Wang, Z., 2024. Aniportrait: Audio-driven synthesis of photorealistic portrait animation. arXiv preprint arXiv:2403.17694.

[3] Bigata, A., Stypułkowski, M., Mira, R., Bounareli, S., Vougioukas, K., Landgraf, Z., Drobyshev, N., Zieba, M., Petridis, S. and Pantic, M., 2025. Keyface: Expressive audio-driven facial animation for long sequences via keyframe interpolation. In Proceedings of the Computer Vision and Pattern Recognition Conference (pp. 5477-5488).

**Questions:**

- On Padding Frames: The paper states, "In previous I2V methods, padding frames were often used for video inference". Does "padding frames" refer to the practice of repeating the initial reference image for several frames to condition the model?

- On the Audio Emotion Module (AEM): Could you clarify the workflow for the AEM? How is the "emotion reference image" selected or provided by the user? Is it an arbitrary image expressing an emotion (e.g., a photo of a smiling person) or a frame from the original video source of the audio?

- Why was this image-based approach chosen over extracting emotional cues directly from the audio signal? This seems to introduce an extra input requirement and limits the emotional expression to be static.

- On Long Video Generation: The appendix clarifies that the long video generation strategy is adapted from Sonic. The wording in the main paper could suggest it is a novel part of your framework. Does this method allow for infinite-length video generation, or is there a practical limit before consistency degrades?

- On the Training Dataset: You mention a curated training set of 500,000 samples. Are there plans to release this dataset or the pipeline used to create it to support reproducibility?

- On the Training Stages: Section 4.1 describes a two-stage training process: first with "audio-only data," then a mix of "audio and image data". What does "audio-only data" mean in this context? Does it imply training without a character reference image, focusing purely on learning a mapping from audio to general motion?

- On the Face-Aware Audio Adapter (FAA): At inference time, is it necessary to provide the model with a face mask for the location of the character in every frame? How is this handled for a sequence of generated frames where the character might move? What are the limitations of this dependency on an external face detection model?

- On Qualitative Comparisons: The figures show excellent results, but would it be possible to provide side-by-side video comparisons with the baseline methods in the supplementary materials? This would greatly strengthen the qualitative evaluation.

---

### Official Review · Reviewer_uKVR · 2025-10-26

**Soundness:** 3
**Presentation:** 3
**Contribution:** 2
**Rating:** 4
**Confidence:** 5

**Summary:**

This paper points out that although significant progress has been made in the field of audio-driven human animation, three core challenges still remain. First, it is difficult to maintain character consistency when generating high-dynamic videos. Second, the accuracy of emotional alignment between characters and audio is insufficient. Third, audio-driven animation for multiple characters cannot be achieved.

This paper proposes: (1) a character image injection module replaces the traditional addition-based character conditioning, eliminating the inherent training-inference condition mismatch and ensuring dynamic motion and strong character consistency. (2) An Audio Emotion Module (AEM) extracts and transfers emotional cues from an emotion reference image to the target generated video, enabling fine-grained, accurate emotion style control. (3) A Face-Aware Audio Adapter (FAA) isolates audio-driven characters with a latent-level face mask, supporting independent audio injection via cross-attention for multi-character scenarios.

**Strengths:**

1.	The generation effect in multi-character scenarios is satisfactory. When controlling Speaker A to speak, Speaker B is not affected by the audio, and there exists a certain degree of eye contact and action interaction between the two speakers. This makes it convincing that the proposed multi-character control method works.
2.	The visual quality and motion diversity are good. This shows that applying this scheme to train the pre-trained DiT model for audio-driven avatar tasks will not cause damage to the DiT performance.
3.	The writing is clear and easy to understand.

**Weaknesses:**

1.	The audio conditional control capability in single-person scenarios seems to be weak. Particularly in the "multi-species" and "multi-style" videos of the main demo, the generated results often suffer from missing syllables or words, even though the speaking rate of the input audio is not particularly fast. As compared in Table 3, I wonder whether CIIS causes a decrease in lip sync.
2.	Although the idea of long video fusion algorithm makes sense to me, there are no long-duration generation results in the supplementary materials. The longest video I could find was only 14 seconds long, which is hardly considered to be a long video.  I am wondering why there are no long demo videos if this technical works well?
3.	Similar to Weakness 2, there are also no emotional controlling demos on the webpage.
4.	See the Ethics Review part. I am confused why this project is called “HunyuanVideo-Avatar” in the abstract of the webpage instead of “AnyAvatar” in the title.

**Questions:**

1.	Discuss and explain the reasons for the weak audio control capability in single-person scenarios.
2.	Explain the reasons for not providing demos of long videos and emotional controlling.
3.	Explain the issues mentioned in the Ethics Review part.

**Details Of Ethics Concerns:**

On the webpage(index.html) of the submitted supplementary materials, the abstract content on the webpage is not consistent with the abstract in the main manuscript. The main differences are the project names, which is called “AnyAvatar”in the main manuscript but also called “HunyuanVideo-Avatar” on the webpage. I am confused and I think the name “HunyuanVideo-Avatar” may reveal some sensitive information about this work.

I am confused and don't exactly know whether this constitutes a violation of the double-blind policy. Therefore, I request an ethics review and verification of this issue of this paper.

---

### Official Review · Reviewer_QFkd · 2025-10-28

**Soundness:** 2
**Presentation:** 2
**Contribution:** 2
**Rating:** 2
**Confidence:** 4

**Summary:**

This work presents a multimodal diffusion transformer for audio-driven human animation that supports dynamic motion, consistent identity, emotion alignment, and multi-character generation. It introduces a character image injection module for dynamic–consistency balance, an Audio Emotion Module for fine-grained emotion control, and a Face-Aware Audio Adapter for multi-speaker scenarios. Extensive experiments demonstrate superior performance over state-of-the-art baselines.

**Strengths:**

The paper presents a modular framework that addresses some challenges in audio-driven human animation. The character image injection module represents a architectural exploration for integrating character appearance into diffusion-based animation. Through ablation studies and comparative experiments (Figure 3, Table 3), the paper provides effectiveness into balancing motion dynamics and identity consistency. Moreover, the videos provided in the supplementary material are of good quality, showcasing vivid and natural facial expressions with high audio–visual synchronization.

**Weaknesses:**

1.  Overall technical contributions are limited, written like a technical experimental report, can't valid the superiority comes from dataset or methods.
2.  The Character Image Injection Module tries to explore an effective way to inject reference images through empirical experiments. However, the paper does not provide sufficiently insightful explanations for why this strategy works, limiting its contribution and the inspiration it offers to readers.
3. The Audio Emotion Module (AEM) is rather naive, representing a straightforward attempt to introduce audio-based emotional control. It lacks careful design to ensure effectiveness — for instance, how will identity-related cues from the emotion reference image severely violating audio conditions? Moreover, the reliance on reference images for emotion control raises concerns about flexibility.
4. The Face-Aware Audio Adapter (FAA) adopts a design that has already been well-discussed in prior works [1][2], with [3] further extending it to a mask-free setting. The paper should include more discussion to clarify its distinctions from these existing approaches and justify its relative advantages.
[1] Hallo3: Highly Dynamic and Realistic Portrait Image Animation with Diffusion Transformer Networks [2] Let Them Talk: Audio-Driven Multi-Person Conversational Video Generation [3] InterActHuman: Multi-Concept Human Animation with Layout-Aligned Audio Conditions

**Questions:**

No questions, please see the weaknesses for rebuttal.

---

### Official Review · Reviewer_tfbk · 2025-10-28

**Soundness:** 3
**Presentation:** 3
**Contribution:** 3
**Rating:** 6
**Confidence:** 4

**Summary:**

This paper proposes a method for audio-driven human animation, addressing limitations in existing approaches related to the trade-off between motion dynamism and visual consistency when using reference images. The authors introduce three main components: (1) a character image injection module to improve consistency without sacrificing motion quality; (2) an Audio Emotion Module (AEM) to align facial expressions with the emotional content of the audio; (3) a Face-Aware Audio Adapter (FAA) to enable independent audio-driven animation of multiple characters by focusing audio influence on specific face regions. The method is evaluated on a newly constructed full-body animation test set containing 250 videos across 200 identities. Objective metrics (including FID, FVD, Sync-C, HKC, HKV, and Q-align) and a user study (assessing lip synchronization, identity preservation, full-body naturalness, and facial naturalness) are used for evaluation. The approach is compared with several state-of-the-art methods for both portrait and full-body animation. Qualitative results show the model's capability in generating diverse character styles, emotion control, and multi-character scenarios.

**Strengths:**

* Improved Character Consistency and Visual Quality: The proposed character image injection module effectively preserves the identity and appearance of the reference character while maintaining dynamic motion, addressing the common trade-off between consistency and motion quality found in previous methods.
* Enhanced Audio-Visual Synchronization for Multiple Characters: The Face-Aware Audio Adapter (FAA) module specifically focuses audio influence on individual face regions, enabling more accurate and independent lip-syncing when animating multiple characters simultaneously from a single audio source.
* Emotion-Aware Facial Animation: The Audio Emotion Module (AEM) leverages emotion labels derived from the input audio to guide facial expression generation, resulting in animations where facial expressions are better aligned with the emotional content of the speech.

**Weaknesses:**

* Subpar quantitative metrics: In the quantitative results presented in Table 1 and Table 2, several metrics for this work are inferior to those of other methods, making it insufficient to fully demonstrate the superiority of the proposed method.
* Reliance on facial masks: The model in this work requires a facial mask, generated by a face detection model, to locate and guide the animation of the head region during both training and inference. While the need for a mask to identify the target ID in multi-character scenarios is understandable, the necessity of a mask in single-character scenarios is unclear. Current mainstream methods (e.g., OmniAvatar[1], FantasyTalking[2], MultiTalk[3], InfiniteTalk[4], etc.) do not require a mask for single-character scenarios, producing results directly from the reference image and audio. Furthermore, adding a facial mask might introduce other issues. It appears that the model in this work only animates the content within the facial mask region. Does this imply that audio information cannot guide the generation of other regions, such as hand gestures or body movements?
* Lack of validation for multi-character driving effectiveness: Several existing methods already support driving multiple people simultaneously on the same screen (e.g., MultiTalk, InfiniteTalk, Playmate2[5], etc.). However, it seems there is currently a lack of appropriate evaluation criteria to assess the performance of this multi-character driving capability.

[1]Gan, Qijun, et al. "OmniAvatar: Efficient Audio-Driven Avatar Video Generation with Adaptive Body Animation." arXiv preprint arXiv:2506.18866 (2025).

[2]Wang, Mengchao, et al. "Fantasytalking: Realistic talking portrait generation via coherent motion synthesis." arXiv preprint arXiv:2504.04842 (2025).

[3]Kong, Zhe, et al. "Let Them Talk: Audio-Driven Multi-Person Conversational Video Generation." arXiv preprint arXiv:2505.22647 (2025).

[4]Yang, Shaoshu, et al. "InfiniteTalk: Audio-driven Video Generation for Sparse-Frame Video Dubbing." arXiv preprint arXiv:2508.14033 (2025).

[5]Ma, Xingpei, et al. "Playmate2: Training-Free Multi-Character Audio-Driven Animation via Diffusion Transformer with Reward Feedback." arXiv preprint arXiv:2510.12089 (2025).

**Questions:**

* The cases presented by this work for multi-character driving scenarios are all limited to two person. Does the method support driving three or more person simultaneously on the same screen?
* Difference from MultiTalk: MultiTalk also supports multi-person driving and enables automatic binding between audio and individuals. What is the fundamental difference between this work's multi-character driving approach and that of MultiTalk? Could this be further discussed?

[1]Kong, Zhe, et al. "Let Them Talk: Audio-Driven Multi-Person Conversational Video Generation." arXiv preprint arXiv:2505.22647 (2025).

---

### Note · Authors · 2025-11-12

**Comment:**

Withdrawal

**Withdrawal Confirmation:**

I have read and agree with the venue's withdrawal policy on behalf of myself and my co-authors.